# Impact of household economic strengthening intervention on food security among caregivers of orphans and vulnerable children in Tanzania

**Amon Exavery**[1]*, **John Charles**[1], **Asheri Barankena**[1], **Shraddha Bajaria**[2], **Epifania Minja**[1], **Jacob Mulikuza**[1], **Tumainiel Mbwambo**[1], **Amal Ally**[1], **Remmy Mseya**[1], **Godfrey M. Mubyazi**[3], **Levina Kikoyo**[1], **Marianna Balampama**[1]

**1** Pact Tanzania, Dar es Salaam, Tanzania, **2** Ifakara Health Institute, Dar es Salaam, Tanzania, **3** National Institute for Medical Research, Dar es Salaam, Tanzania

* aexavery@pactworld.org

**Data Availability Statement:** All relevant data are within the paper and its Supporting information files.

## Abstract

About 2 billion people worldwide suffer moderate or severe forms of food insecurity, calling for correctional measures involving economic strengthening interventions. This study assessed the impact of household economic strengthening (HES) intervention on food security among caregivers of orphans and vulnerable children (OVC) in Tanzania. The study was longitudinal in design, based on OVC caregivers' baseline (2017–2018) and midline (2019) data from the USAID Kizazi Kipya project. Food security, the outcome, was measured using the Household Hunger Scale (HHS) in three categories: little to no hunger (food secure), moderate hunger, and severe hunger. Membership in the USAID Kizazi Kipya-supported economic strengthening intervention (i.e. WORTH Yetu) was the main independent variable. Data analysis involved generalized estimating equation (GEE) for multivariate analysis. With mean age of 50.3 years at baseline, the study analyzed 132,583 caregivers, 72.2% of whom were female. At midline, 7.6% of all caregivers enrolled at baseline were members in WORTH Yetu. Membership in WORTH Yetu was significantly effective in reducing household hunger among the caregivers: severe hunger dropped from 9.4% at baseline to 4.1% at midline; moderate hunger dropped from 65.9% at baseline to 62.8% at midline; and food security (i.e., little to no hunger households) increased from 25.2% at baseline to 33.1% at midline. In the multivariate analysis, membership in WORTH Yetu reduced the likelihood of severe hunger by 47% (OR = 0.53, 95% CI 0.48–0.59), and moderate hunger by 21% (OR = 0.79, 95% CI 0.76–0.83), but increased the likelihood of food security by 45% (OR = 1.45, 95% CI 1.39–1.51). The USAID Kizazi Kipya's model of household economic strengthening for OVC caregivers was effective in improving food security and reducing household hunger in Tanzania. This underscores the need to expand WORTH Yetu coverage. Meanwhile, these results indicate a potential of applying the intervention in similar settings to address household hunger.

**Funding:** The USAID Kizazi Kipya project (2016 – 2021) is funded by the President's Emergency Plan for AIDS Relief (PEPFAR) through the United States Agency for International Development (USAID) and it is being implemented by Pact in Tanzania. The contents of this paper, the design of the study, collection, analysis, and interpretation of data as well as writing of the manuscript remain the sole responsibility of the authors, and do not necessarily reflect the views of USAID or the United States Government. The funder provided support in the form of salaries for authors [AE, JC, AB, EM, JM, TM, AA, RM, LK, and MB] who are Pact employees, but did not have any additional role.

**Competing interests:** The authors have declared that no competing interests exist.

## Introduction

The Food and Agriculture Organization (FAO) estimates that 2 billion people of the world's population are experiencing moderate or severe food insecurity [1]. Primarily concentrated in low- and middle-income countries (LMICs), these people are facing irregular access to nutritious and sufficient food. Because of this, they are exposed to a greater risk of malnutrition and poor health in general [1]. The situation is further compounded by the COVID-19 pandemic [2], largely exerting considerable challenges to governments and their allied development partners in their strive for attaining the United Nations' (UN) Sustainable Development Goal (SDG) of Zero Hunger by 2030. The UN believes that the SDGs are not possible to achieve without ending hunger and malnutrition, and without having sustainable and resilient agriculture and food systems [3].

Food insecurity has been associated with several adverse outcomes, including among others, an increased HIV transmission risk [4, 5] through transactional sex and inability to negotiate safer sex [4], poor clinic attendance [6], poor antiretroviral therapy (ART) uptake and adherence [4, 6, 7], lower likelihood of achieving complete viral suppression [8], poor immunological and virological responses, lower efficacy of ART, and high mortality [6, 8].

Potentially successful interventions to address food insecurity have been proposed: nutrition supplementation for people living with HIV (PLHIV), food and cash transfers to affected households, and livelihood programs for individuals affected by long-term food insecurity [6]. However, underlying causes of food insecurity, especially climate change, conflicts and security issues cannot be ignored in areas where they count [6].

Poverty and food insecurity are linked [9–11]. Poverty constitutes a major exposure to different forms of vulnerabilities [12, 13], including poor health, limited access to education and hunger, among others [13]. The consequences of poverty are far-reaching, with profound pernicious effects mostly felt by children, especially orphans. Children and young people represent majority of sub–Saharan Africa's populations. However, millions of these are orphans [14], largely because of the high adult mortality due to HIV and AIDS in the region [15]. As HIV and AIDS alters family structures due to deaths of family members, older persons become responsible for providing care (i.e., caregivers) to orphaned children. Therefore, the number of grandparents caring for AIDS orphans in developing countries has doubled over the last decade [16]. Majority of the caregivers are women who face serious financial, physical and emotional stress due to caregiving obligations [16]. These caregivers are more likely to experience hunger due to additional economic pressure in caring for OVC [17]. Studies have shown that orphaned children are more likely than their non-orphaned counterparts to grow up in poverty, drop out of school, and contract HIV [14].

Although trends are declining, poverty remains a significant problem in Tanzania. According to the 2017–18 Household Budget Survey (HBS), about a quarter (26.4%) of the Tanzania's population are experiencing basic needs poverty, and 8.0% are extremely poor as they live below food poverty line [18]. The situation may be even worse for households caring for orphans and vulnerable children (OVC), therefore underscoring the need for immediate poverty alleviation interventions and strategies to rescue these families. In view of this, household economic strengthening (HES) interventions have been recommended to address economic vulnerabilities and empower families to meet essential needs of the children and everyone in the household. HES interventions such as individual and group savings, financial education, cash transfers, income generation and vocational training have the potential to address structural barriers, particularly poverty [19–22]. The main feature of savings groups is provision of interest-bearing loans and each member determines a personal amount of money to contribute [23].

The literature on the impacts of HES interventions on food security is apparently sporadic, and limited in terms of sample size, geographical coverage, and depth of statistical analyses. This study contributes to the discourse by assessing the impact of household-level economic strengthening (HES) interventions on household food security among OVC caregivers in Tanzania. The caregivers are taking care of children whose orphanhood or vulnerabilities are HIV-related [24]. In this process, they are also vulnerable and more likely to experience food insecurity because of poverty attributable to additional burden in caring for OVC.

## Materials and methods

### Data source

Data for this study stem from the USAID Kizazi Kipya project (2016–2021) in Tanzania. The project aims at scaling up the uptake of HIV services, and other health and social services by OVC and their caregivers. Data were collected by Lead Case Workers (LCWs) and Community Case Workers (CCWs) during beneficiary screening and enrollment using the project's screening and enrollment, and Family and Child Asset Assessment (FCAA) tools. LCWs and CCWs are lay social welfare volunteers recruited by a government standard and trained in basic social welfare case management skills. Beneficiaries were enrolled in the project if their household met one or more of the enrollment criteria, referring to fourteen household vulnerabilities related to HIV [24]. The enrollment data were collected during 2017–2019 and were referred to as baseline. After enrollment, service provision followed, including services for household economic strengthening (HES). In 2019, a follow-up data collection using the same tools and procedures was conducted among the same caregivers and was referred to as midline.

### Household economic strengthening under the USAID Kizazi Kipya project

As highlighted above, the USAID Kizazi Kipya project serves OVC and their caregivers. The project is community-based and provides need-based services that are integrated, covering several dimensions, particularly, HIV services, other health services, food and nutrition, psychosocial care and support, and HES. Details of various services offered by the USAID Kizazi Kipya project can be accessed [25].

HES intervention under the project is aimed at increasing the financial capacity of the households in meeting their basic needs. The intervention is organized through WORTH Yetu groups which are locally formed and comprising the project and non-project beneficiaries. The intention of having mixed beneficiaries is to reduce stigma to HIV positive household members and to promote inclusivity in the community. The WORTH Yetu groups require members to have weekly mandatory and voluntary savings that form the base for individual loans and group startups projects.

The WORTH Yetu groups have three sets of collection funds, namely, social fund, OVC fund, and community resource mobilization fund. All these collection funds contribute to food security in the households by supporting households that are malnourished, child headed and those that are not capable of having three meals per day. The respective households are supported through food packages and are given loans to start business to support the household. The loans are given to only WORTH Yetu group members so that they can adhere to the repayment plan and schemes provided. To meaningfully contribute to food security and reduction of household poverty, the majority of the WORTH Yetu groups have started group projects such as farming, animal husbandry and horticulture.

The project views economic strengthening as a pathway towards growth and reduction of OVC households' vulnerability. Economic vulnerabilities of households are identified through

administration of Family and Child Asset Assessment (FCAA) which categorizes OVC households into four sequential HES: provision, protection, production, and promotion. The pathway sees OVC households progressing sequentially from provision through promotion. The entry point in the pathway depends on the initial status of the households at enrollment. The rate of progression depends on household members' capacity, orientation, and motivations [26].

Destitute households in provision category are linked to direct provisional support to meet basic needs and recover key assets they have lost. The project provides health insurance to these households and facilitates linkage to provisional service providers, including linkage to WORTH Yetu groups where OVC fund and community resource mobilization funds are specifically set to address needs of OVC. Households in protection category tend to prioritize improvements in their ability to manage consumption and cash flow over improvements in their income. These are mobilized to establish WORTH Yetu savings and lending groups as basic self-insurance mechanisms to strengthen their loss-management strategies and improve the way they manage household risks. Through WORTH Yetu groups, OVC caregivers get access to financial literacy, opportunity to save and access to micro credits. Households in production category are ready and willing to invest in risk-sensitive income generating activities. At this stage, the project facilitates linkage to extension services and input suppliers to facilitate establishment and scaling of income generating activities. The project further invests in market assessment studies to enable informed investment decisions among beneficiaries. Households in promotion category have solid self-insurance mechanisms plus reliable and frequent income streams to allow them to make riskier investments in higher-return economic activities. The project provides these households with tailor-made support to facilitate linkages to credits and markets for their products.

## Study area

Data for this study originate from 81 district councils in 25 regions of Tanzania where the USAID Kizazi Kipya project had implemented screening and enrollment activities in 2017–2018. The respective regions were: Arusha, Dar es Salaam, Dodoma, Geita, Iringa, Kagera, Katavi, Kigoma, Kilimanjaro, Mara, Mbeya, Mjini Magharibi, Morogoro, Mtwara, Mwanza, Njombe, Pwani, Rukwa, Ruvuma, Shinyanga, Simiyu, Singida, Songwe, Tabora, and Tanga.

## Study design and data collection tools

This was a longitudinal study [27], whereby data used herein were collected at enrollment as baseline from 16th March 2017 to 18th April 2019. After enrollment, service delivery to all enrolled beneficiaries followed with regular household visits by LCWs and CCWs. From 1st February to 30th September 2019, another assessment referred to as midline was conducted using the FCAA tool. From baseline (i.e. enrollment) to midline, the beneficiaries had been in the project for a duration ranging from 0.01 to 2.50 years depending on their enrollment date. The FCAA tool captured caregivers' demographic information, household assets, sources of income, HIV status, food security, and use of and adherence to antiretroviral therapy (ART) for those who reported their HIV status as positive.

## Study population

The current study was based on 132,583 caregivers of OVC from 132,583 households. These were beneficiaries of the USAID Kizazi Kipya project from 16th March 2017 to 30th September 2019. During this period, each of the caregivers was assessed twice (i.e. at baseline and at midline) using the FCAA tool. Caregivers who were not assessed at the midline were not included in the current study. A caregiver is defined by the USAID Kizazi Kipya project as a

guardian who has the greatest responsibility for the daily care and rearing of one or more OVC in a household. A caregiver is not necessarily a biological parent of the OVC. Therefore, in each enrolled household, one caregiver was registered in the project, making the number of caregivers and that of households equal.

## Variables

Food security which was measured by assessing the level of household hunger was the dependent (i.e., outcome) variable, objectively measured using the Household Hunger Scale (HHS) designed by FAO and the Tufts University through the Food and Nutrition Technical Assistance III Project (FANTA) [28].

The HHS originates from the Household Food Insecurity Access Scale (HFIAS) [28]. It is a validated scale for cross-cultural use in food insecure areas, thus different from other food security indicators [28]. The scale operationalizes three main questions which are administered during field data collection: (a) in the past 4 weeks, how often was there ever no food to eat of any kind in your household because of lack of resources to get food?, (b) in the past 4 weeks, how often did any household member go to sleep at night hungry because there was no enough food?, and (c) in the past 4 weeks, how often did any household member go whole day and night without eating anything? All these questions were inclusive in the data collections tools (i.e. FCAA) used by the USAID Kizazi Kipya project to collect both baseline and midline data.

According to the HHS, household hunger is classified in three groups based on household hunger score, whereby, the bigger the score the hungrier the household. Each of these questions has a set of four (4) possible responses: never, rarely (once or twice), sometimes (3 to 10 times), and often (more than 10 times). In the computation process, the first category, never, is recoded as '0', the next two categories, rarely (once or twice) and sometimes (3 to 10 times) are recoded as '1', and the last category, often (more than 10 times), is recorded as '2'. Then for each household, total score across the three questions is computed. Possible values for the resulting scores range from zero (0) to six (6), which are finally organized in the hunger categories as: (a) little to no hunger (score 0–1), (b) moderate hunger (score 2–3), and (c) severe hunger (score 4–6) [28].

The main independent variable for the current study was participation in the USAID Kizazi Kipya-supported economic strengthening intervention, WORTH Yetu. The WORH Yetu refers to savings and loan groups that put together savings deposits into a fund from which loans are issued to group members. This is a platform for members to gain financial literacy, business skills, and financial management among others. For the current study, caregivers' unique identification numbers were used to identify those who were WORTH Yetu members from the USAID Kizazi Kipya's WORH Yetu database.

Several other independent variables were included in the study: caregiver sex, age, education, marital status, whether the caregiver was mentally or physically disabled, whether the caregiver had health insurance, type of residence (rural or urban). Rural residence included all those living in district councils, whereas those living in townships, municipal or city councils were considered as urban residents.

## Data analysis

All statistical analyses were performed using Stata 14.0 statistical software. Distributional features of the caregivers were obtained through one-way tabulations. For both baseline and midline, the association between food security and each of the independent variables was tested using the Chi-Square ($\chi^2$) test because all variables were categorical.

In the multivariate analysis, three models were fitted. Each of the food security categories (i.e. little to no hunger, moderate hunger, and severe hunger) was treated as a separate outcome

variable, coded as "1" for caregivers belonging in the category and "0" otherwise. This approach was applied in another study [29]. The multivariate analysis was thus conducted using generalized estimating equation (GEE) with logit link function, binomial distribution family and an unstructured correlation structure. Since food security was measured twice (i.e. at baseline and at midline) for each caregiver, there were 132,583*2 = 265,166 total observations for which each multivariate model was conducted. Therefore, baseline and midline observations of the same caregiver were assumed to be correlated. This made the GEE the most appropriate model because the model addresses within-subject correlations and both time-dependent and time-independent covariates [30–35]. The model took the following form:

$$Y_{it} = \beta_0 + \sum_{j=1}^{J} \beta_{1j} X_{itj} + \beta_2 t + \cdots + CORR_{it} + \varepsilon_{it}$$

Where $Y_{it}$ are food security observations for caregiver i at time t, $\beta_0$ is the intercept, $X_{ijt}$ is the independent variable j for caregiver i at time t, $\beta_{1j}$ is the regression coefficient for independent variable j, J is the number of independent variables, t is time, $\beta_2$ is the regression coefficient for time, $CORR_{it}$ is the working correlation structure (unstructured in this case), and $\varepsilon_{it}$ is the 'error' term for caregiver i at time t. Food security was measured at two time points: at baseline ($t_0$) where caregivers were being enrolled into the project and had not received any services from the USAID Kizazi Kipya project; and at the midline ($t_1$) where caregivers had received different program services, including economic strengthening.

Furthermore, the multivariate analysis for this study was guided by a theory-informed conceptual framework (Fig 1) developed based on the World Food Programme (WFP)'s Food and Nutrition Security Conceptual Framework [36] and Townsend et al.'s conceptual framework of food insecurity and its relation to overweight [37]. According to WFP, there are multiple levels that influence food security: individual, households, and community. Accordingly, food security is linked to political, institutional, and environmental dynamics [36]. These levels reflect the four dimensions of food insecurity suggested by Townsend and colleagues as demographic characteristics (e.g. age, ethnicity etc.), socioeconomic status (e.g. education, income, occupation etc.), government assistance (e.g. welfare, food stamps etc.), and environment (e.g. region, urbanization etc.) [37]. Applied to this study, the effect of membership in WORTH Yetu as the main explanatory variable on household hunger (the outcome) was regressed in multivariate GEE models, taking into account the demographic characteristics (i.e. sex, age, marital status), socioeconomic status (i.e. education), individual (i.e. mental or physical disability status, HIV status), household (i.e. health insurance), and environmental characteristics (i.e. place of residence). These variables were included in the model to ensure that they do not confound the relationship between WORTH Yetu membership and household hunger. This approach has been applied in similar studies [37, 38].

Apart from WORTH Yetu which was the main independent variable, the rest of the independent variables were included in the multivariate models one at a time and retained only if log likelihood ratio test showed that their presence improved the overall model [39]. In the model specification, coefficients were exponentiated using Stata's '*eform*' option to obtain adjusted odds ratios (ORs) and their corresponding 95% confidence intervals (CIs). All statistical inferences were made at a significance level of 5% ($\alpha = 0.05$).

## Ethics approval

Ethics approval for this study was received from the Medical Research Coordinating Committee (MRCC) of the National Institute for Medical Research (NIMR) in Tanzania with certificate number NIMR/HQ/R.8a/Vol.IX/3024. Recruitment of beneficiaries into the USAID

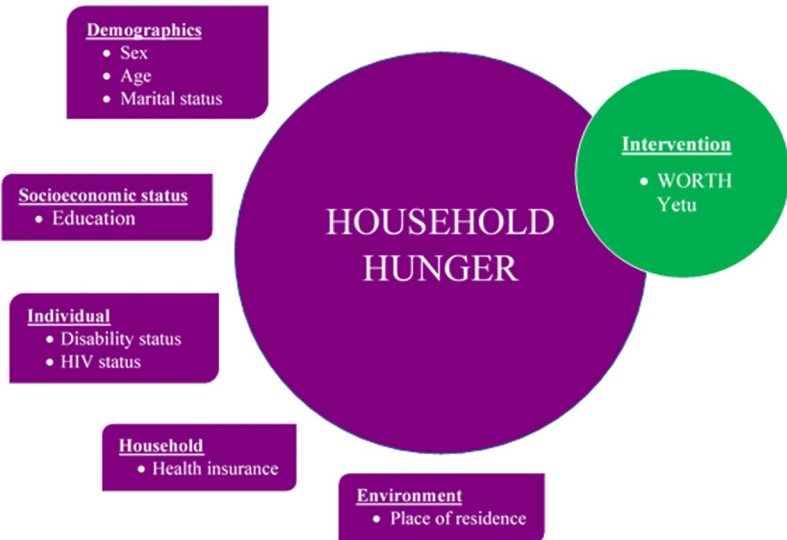

**Fig 1. Conceptual framework of the effect of WORTH Yetu intervention on household hunger, considering demographic, socioeconomic, individual, household, and environmental characteristics of OVC caregivers in Tanzania.**

Kizazi Kipya program was voluntary. All households which were identified as eligible for the program per the program enrollment criteria, and accepted to be enrolled in the program, signed an informed consent form after which the enrollment followed. Information of project beneficiaries is stored very securely and confidentially, and datasets analyzed for the current study were anonymous.

## Results

### Profile of respondents

This study was based on 132,583 caregivers of OVC in Tanzania. At baseline, the caregivers were aged 50.3 years on average (with standard deviation of 14.7 years). The youngest and oldest caregiver was aged 18 and 100 years, respectively. Majority of the caregivers were female (72.2%) and had primary education (73.7%). Most of the caregivers were either married or living together with their spouses (45.1%) and widowed (32.7%). Slightly more than a half of the caregivers lived in rural areas (55.2%), the rest lived in urban areas. Four percent (4.0%) of the caregivers were mentally or physically disabled, 13.4% had health insurance, and 37.7% were living with HIV (Table 1).

In addition, from the baseline (i.e. at enrollment) to the midline, the caregivers had received services from the USAID Kizazi Kipya project for different durations depending on their enrollment dates, ranging from 0.01 to 2.50 years (mean = 1.40, standard deviation (SD) = 0.51) (Fig 2).

### Membership in WORH Yetu

At baseline (i.e., at enrollment), there were 0% of the 132,583 caregivers in USAID Kizazi Kipya–supported HES intervention (i.e., WORTH Yetu). Later, at midline, 7.6% (n = 10,123) of the caregivers had become members of WORTH Yetu.

**Table 1. Caregivers' baseline characteristics (n = 132,583).**

| | Number of respondents (n) | Percent (%) |
|---|---:|---:|
| **ALL** | **132,583** | **100.0** |
| **Sex** | | |
| Female | 95,661 | 72.2 |
| Male | 36,922 | 27.9 |
| **Age (in years)** | | |
| 18–29 | 6,798 | 5.1 |
| 30–39 | 26,109 | 19.7 |
| 40–49 | 38,454 | 29.0 |
| 50–59 | 25,217 | 19.0 |
| 60+ | 36,005 | 27.2 |
| Mean = 50.3, SD = 14.7, min. = 18, Max. = 100 | — | — |
| **Marital status** | | |
| Married or living together | 59,770 | 45.1 |
| Divorced or separated | 19,576 | 14.8 |
| Never married | 9,856 | 7.4 |
| Widow/Widower | 43,381 | 32.7 |
| **Education** | | |
| Never attended | 30,508 | 23.0 |
| Primary | 97,767 | 73.7 |
| Secondary+ | 4,308 | 3.2 |
| **Place of residence** | | |
| Rural | 73,141 | 55.2 |
| Urban | 59,442 | 44.8 |
| **Mentally or physically disabled?** | | |
| No | 127,253 | 96.0 |
| Yes | 5,330 | 4.0 |
| **Has health insurance?** | | |
| No | 114,815 | 86.6 |
| Yes | 17,768 | 13.4 |
| **HIV status** | | |
| Negative | 47,251 | 35.6 |
| Positive | 49,970 | 37.7 |
| Undisclosed/unknown | 35,362 | 26.7 |

## Food security at baseline and midline

As shown in Fig 3, at baseline, 25.2% of the caregivers were in little to no hunger households, 65.4% in moderate hunger households, and 9.4% in severe hunger households. This pattern was seen at the midline, but with significant changes in the estimates, whereby, 33.1% of the caregivers were in little to no hunger households, 62.8% in moderate hunger households, and 4.1% in severe hunger households at the midline. This reflected a significant improvement ($p < 0.001$) in household food security between baseline and midline. Caregivers in little to no hunger households (i.e. food secure) increased by 7.9%, from 25.2% at baseline to 33.1% at the midline. Caregivers in moderate hunger households declined by 2.6%, from 65.4% at baseline to 62.8% at the midline. The most decline was 5.3%, observed among caregivers in severe hunger households from 9.4% at baseline to 4.1% at the midline.

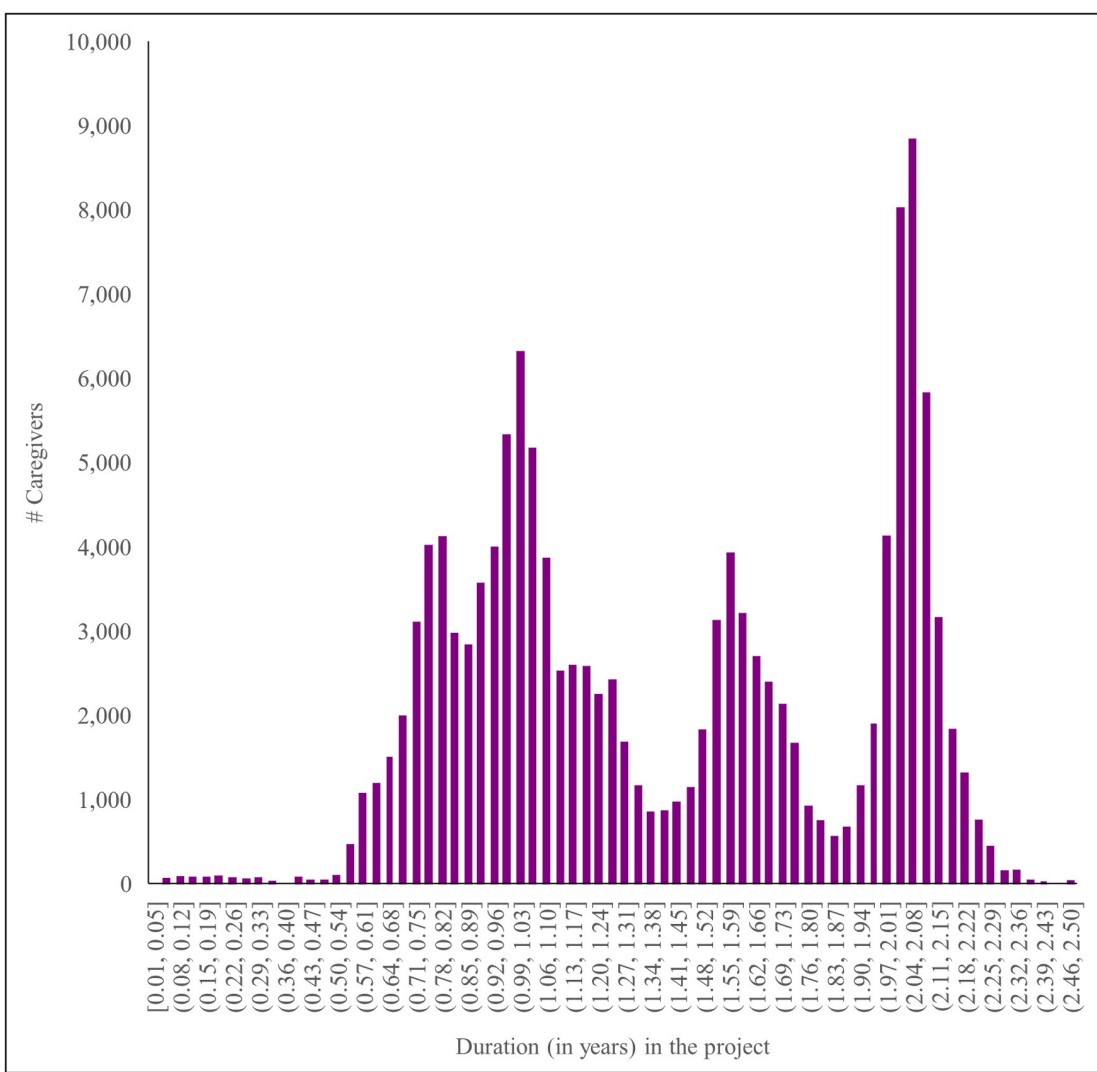

**Fig 2. Caregivers' duration (in years) in the USAID Kizazi Kipya project from baseline (enrollment) to midline (n = 132,385).**

## Food security at midline by membership in WORTH Yetu

At midline, overall, 33.1%, 62.8% and 4.1% of the caregivers were in households with little to no hunger, moderate hunger, and severe hunger, respectively. These proportions varied significantly by WORTH Yetu membership status. While 41.3% of the caregivers in WORTH Yetu

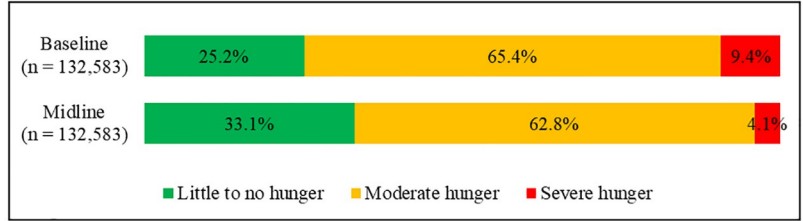

**Fig 3. Caregivers' level of household hunger at baseline and midline (n = 132,583).**

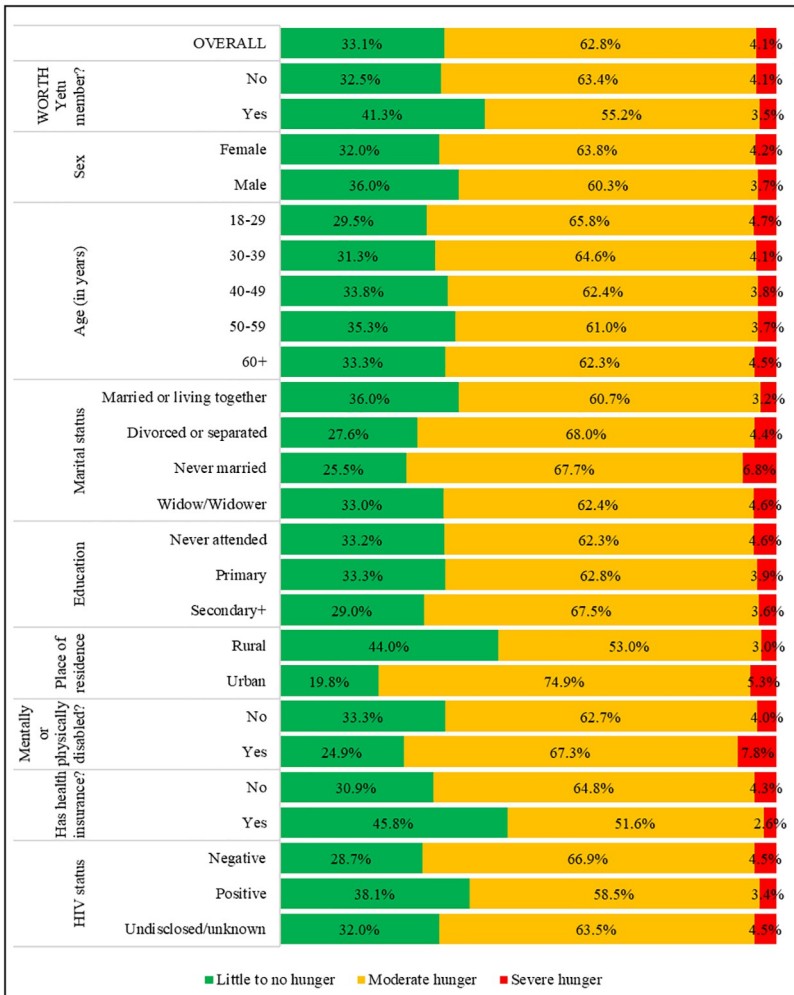

**Fig 4. Caregivers' level of household hunger at midline by background characteristics (n = 132,583).**

were in little to no hunger households at the midline, so was 32.5% among nonmembers. Also, members of WORH Yetu were significantly declining in subsequent higher levels of household hunger compared to nonmembers: moderate hunger: 55.2% vs. 63.4%; and severe hunger: 3.5% vs. 4.1% (Fig 4).

## Food security at midline by background characteristics

Fig 4 further shows the extent of food security at midline by caregiver's background characteristics. There were more male than female caregivers in the little to no hunger households, but female caregivers exceeded their male counterparts in both moderate and severe hunger households (p<0.001). With respect to age, the proportion of caregivers in the little to no hunger households increased with age, except age group 60+ years whose proportion slightly declined, although it remained above that of the youngest two age categories.

Accordingly, both the moderate and severe hunger household declined with age, except the oldest age category which showed differences (p<0.001). The proportion of caregivers in little to no hunger households was highest among those who were married or living together with their spouses and lowest among the never married ones. But, while the moderate hunger was

highest among the caregivers who were divorced or separated, severe hunger was highest among those who were never married, and both were lowest among those were married or living together with their spouses (p<0.001). There were more caregivers in the moderate to no hunger households in the rural areas than in urban ones. Conversely, caregivers in both moderate and severe hunger households were higher in urban than in rural areas (p<0.001). Other factors which were associated with food security in the cross tabulations were education (<0.001), mental or physical disability (p<0.001), health insurance (p<0.001), and HIV status (p<0.001).

## Results of multivariate analysis

Table 2 contains three models (Model 1, Model 2, and Model 3) each of which shows adjusted odds ratios (ORs) and their corresponding 95% confidence intervals (CIs) of multivariate GEE of the effect of WORTH Yetu and other covariates on household hunger. Results are interpreted considering that the effect of WORTH Yetu on household hunger is adjusted for caregiver sex, age, marital status, education, place of residence, mental or physical disability, health insurance, and HIV status.

As the results reveal, the caregivers who were members in WORTH Yetu were 45% more likely to be in little to no hunger households than non-members (OR = 1.45, 95% CI 1.39–1.51) (Model 1). In Models 2 and 3, WORTH Yetu members were 21% and 47% less likely to be in moderate hunger households (OR = 0.79, 95% CI 0.76–0.83) and severe hunger households (OR = 0.53, 95% CI 0.48–0.59) compared to the non-members, respectively. Overall, these observations suggest that participation in WORTH Yetu was protective against household hunger, and non-participation increased household hunger.

Other factors which were significantly associated with household hunger included sex, whereby, male caregivers were more likely to be in severe hunger households than their female counterparts. Regarding age, food security improved as age increased. Also, as age increased, the likelihood of belonging in moderate hunger households, and severe hunger households declined in both cases.

With respect to marital status, all the three models in Table 2 support that those caregivers in marital categories other than married or living together were less likely to be in the little to no hunger households, but more likely to experience moderate hunger, and severe hunger.

Additionally, all the three models in Table 2 show that both the caregivers who had primary education, and secondary and above were more likely to be in the little to no hunger households, but less likely to be in moderate hunger households, and severe hunger households than those who had never been to school. While caregivers in urban areas were less likely to be in little to no hunger households, they were more likely to be in both moderate and severe hunger households, than their rural counterparts. Similarly, physically or mentally disabled caregivers were less likely to be in little to no hunger households, but more likely to be in both moderate and severe hunger households than their non-disabled counterparts. Caregivers with health insurance were more likely to be in little to no hunger households, but less likely to be in moderate, and severe hunger households than those without. HIV status was associated with household hunger, in such a way that, the caregivers who were LHIV and those who did not disclose their status were more likely to be in little to no hunger, and severe hunger households, but less likely to be in moderate hunger households than those who were HIV negative.

## Discussion

This study assessed the contribution of the USAID Kizazi Kipya project's household economic strengthening intervention on food security among OVC caregivers in Tanzania. Overall,

**Table 2. Multivariate generalized estimating equations (GEE) of the role of membership in WORTH Yetu on household level of hunger among caregivers of OVC in Tanzania.**

| | Household level of hunger | | | | | |
| --- | --- | --- | --- | --- | --- | --- |
| | **(Model 1) Little to no hunger (n = 265,166)** | | **(Model 2) Moderate hunger (n = 265,166)** | | **(Model 3) Severe hunger (n = 265,166)** | |
| Covariate | **Odds Ratio (OR)** | **95% Confidence Interval (CI)** | **Odds Ratio (OR)** | **95% Confidence Interval (CI)** | **Odds Ratio (OR)** | **95% Confidence Interval (CI)** |
| **WORH Yetu member?** | | | | | | |
| No | 1.00 | — | 1.00 | — | 1.00 | — |
| Yes | 1.45*** | 1.39, 1.51 | 0.79*** | 0.76, 0.83 | 0.53*** | 0.48, 0.59 |
| **Sex** | | | | | | |
| Female | 1.00 | — | 1.00 | — | | |
| Male | 0.98 | 0.96, 1.004 | 0.99 | 0.97, 1.01 | 1.11*** | 1.07, 1.15 |
| **Age (in years)** | | | | | | |
| 18–29 | 1.00 | — | 1.00 | — | 1.00 | — |
| 30–39 | 1.04 | 0.99, 1.09 | 0.99 | 0.95, 1.03 | 0.94 | 0.87, 1.01 |
| 40–49 | 1.06** | 1.02, 1.11 | 0.98 | 0.94, 1.02 | 0.88** | 0.82, 0.95 |
| 50–59 | 1.11*** | 1.06, 1.16 | 0.94** | 0.91, 0.98 | 0.88** | 0.82, 0.95 |
| 60+ | 1.13*** | 1.08, 1.18 | 0.93** | 0.90, 0.97 | 0.86*** | 0.80, 0.92 |
| **Marital status** | | | | | | |
| Married or living together | 1.00 | — | 1.00 | — | 1.00 | — |
| Divorced or separated | 0.77*** | 0.75, 0.79 | 1.20*** | 1.17, 1.23 | 1.22*** | 1.17, 1.28 |
| Never married | 0.79*** | 0.76, 0.82 | 1.08*** | 1.05, 1.12 | 1.46*** | 1.38, 1.55 |
| Widow/Widower | 0.89*** | 0.87, 0.91 | 1.05*** | 1.03, 1.07 | 1.22*** | 1.17, 1.26 |
| **Education** | | | | | | |
| Never attended | 1.00 | — | 1.00 | — | 1.00 | — |
| Primary | 1.22*** | 1.19, 1.25 | 0.90*** | 0.88, 0.92 | 0.77*** | 0.74, 0.79 |
| Secondary+ | 1.47*** | 1.39, 1.56 | 0.79*** | 0.75, 0.84 | 0.70*** | 0.64, 0.77 |
| **Place of residence** | | | | | | |
| Rural | 1.00 | — | 1.00 | — | 1.00 | — |
| Urban | 0.38*** | 0.37, 0.39 | 2.20*** | 2.16, 2.24 | 1.22*** | 1.18, 1.26 |
| **Mentally or physically disabled?** | | | | | | |
| No | 1.00 | — | 1.00 | — | 1.00 | — |
| Yes | 0.68*** | 0.64, 0.72 | 1.18*** | 1.12, 1.24 | 1.54*** | 1.44, 1.66 |
| **Has health insurance?** | | | | | | |
| No | 1.00 | — | 1.00 | — | 1.00 | — |
| Yes | 1.53*** | 1.50, 1.57 | 0.70*** | 0.69, 0.72 | 0.80*** | 0.76, 0.84 |
| **HIV status** | | | | | | |
| Negative | 1.00 | — | 1.00 | — | 1.00 | — |
| Positive | 1.26*** | 1.23, 1.28 | 0.81*** | 0.79, 0.83 | 1.04** | 1.01, 1.09 |
| Undisclosed/Unknown | 1.04** | 1.02, 1.07 | 0.94*** | 0.92, 0.96 | 1.12*** | 1.08, 1.16 |

Significance:

***p<0.001,

**p<0.05;

For each of the three models: number of caregivers = 132,583, number of observations per caregiver = 2, total number of observations analyzed for each model = 265,166.

findings reveal that there was a significant improvement in household food security among the caregivers by 7.9%, from 25.2% at baseline to 33.1% at midline. Accordingly, household hunger declined significantly: moderate hunger households fell by 2.6%, from 65.4% at baseline to 62.8% at midline, and severe hunger households dropped by 5.3%, from 9.4% at baseline to 4.1% at midline. Indeed, multivariate analysis confirmed the observations, with participation in WORTH Yetu being the underlying driver. As noted, the participants of WORTH Yetu were 45% more likely to be in little to no hunger households than non-participants. On the other hand, participants of the WORTH Yetu were less likely by 21% and 47% to be in moderate hunger households, and severe hunger households, respectively, than non-participants. These findings were consistent with others showing positive impacts in food security attributable to participation in savings groups in Uganda [40], Malawi [41] and elsewhere [42]. Similarly, a food security intervention in Rwanda resulted in significant improvement in food accessibility [43].

Further to this, a study by Kwilasa [44] remarked that, "*...compared to non-participants, households particularly women that have access to credit and savings enjoy a higher food security than their counterparts. More women have access to resources, and they are using them buy inputs, either through the group share out scheduled to coincide with agricultural seasons, or by getting a loans or credit from the VSLAs. There is progress in reduction in malnutrition rates of communities that have benefited from food security training and activities. This is particularly the case in VSLA households, where women report that their children are better nourished and where there seems to be fewer referral cases to health centers due to malnutrition...*"

Defined by the Word Health Organization (WHO) as non-medical factors that influence health outcomes [45], social determinants of health (SDH) include food insecurity, a factor which has been identified to be having one of the most extensive impacts on the overall health of individuals [46]. As a result, SDH account for up to 55% of the global health outcomes, making the non-health sectors to contribute more to population health outcomes than the health sector [45]. Therefore, building food security programs stands out to be one of the core livelihood strategies to tackle SDH [47]. One of the strategies is HES, an intervention that this study tested and found to be effective in addressing food insecurity among vulnerable populations in Tanzania. HES has also been identified as a poverty alleviation strategy in other populations and settings [19–22].

Findings of the present study imply that, the USAID Kizazi Kipya's model of household economic strengthening represents a working solution to addressing economic vulnerabilities among poor families in resource-poor settings. At the program level, the challenge seems to be with coverage since only 7.6% of the caregivers analyzed were members of WORH Yetu at midline. This could be partly considered a result of the WORTH Yetu intervention not being available right after the baseline data were collected. Thus, omission of membership barriers should be of paramount priority towards reaching all caregivers with the WORTH Yetu intervention to structurally enable them to meet an important need.

However, this study uncovered other factors with significant association with food security among the caregivers. The study found out that as the caregiver age advanced, the odds of belonging in little to no hunger households increased, and this was supported by the declining odds of belonging in both moderate and severe hunger households with age. This was an expected finding and indeed consistent with observations reported from other studies [48–51]. A possible mechanism for this observation is that household income may improve and consequently improve food security as a result of caregiver's longer stay on an economic endeavor or income-generating activities. Also, to enhance food security as an outcome of livelihood strategies [52], reasonable time may be required for positive gains to upshot, hence a possible pathway of the observed higher likelihood of improved food security among older caregivers

than their younger counterparts. Therefore, younger caregivers (<40 years) may require targeted economic strengthening support to improve their food security situation.

Also, although there were similar likelihoods between male and female caregivers of being in both little to no hunger, and moderate hunger households; male caregivers were significantly more likely to be in severe hunger households than their female counterparts. This validates what has been reported from similar studies that found female gender's positive impact on food security [53, 54]. In connection to this, it has been argued that in times of economic distress, men tend to isolate themselves from others and take individual actions while women more often engage their social network for support [54]. In this case, men's economic status may further deteriorate and consequently bump into severe hunger, unlike women. This observation underscores the need for the government, development partners, and programs to design and deliver food security interventions with additional support for men, especially male caregivers' households, as they may be likely to suffer the most acute form of food insecurity.

Further, it was noted that, caregivers who were divorced or separated, never married (i.e., single), and widowed were all less likely to be food secure, hence more likely to experience both moderate and severe hunger than those who were married or living together with their spouses. It could be that outside marital unions, caregivers were more likely to experience food insecurity possibly due to lack of social support. This is because, social support has been observed in another study as a mediator of economic hardships [54]. Therefore, the findings of the current study suggest that delivery of economic strengthening interventions should be tailored to marital status to address marital-related situations which affect participation in HES and other interventions for improved food security.

With respect to education, the study observed, in a dose–response fashion, that the higher the education the higher was the likelihood of belonging in little to no hunger households (i.e. being food secure). This is consistent with several studies [48, 50, 55, 56]. Education provides knowledge and awareness and increases the likelihood of having good jobs and or businesses, hence better income which is key to food security. It is also possible that caregivers with primary education and beyond were more likely to comprehend and apply financial literacy which is also a key component in the USAID Kizazi Kipya's economic strengthening intervention. As one of the key SDH [45], formal education attainment should be encouraged on one hand, and target HES and food security interventions to caregivers who have never attained formal education, on the other. The latter is more important than the former because as adults with OVC and other family members to take care of, the caregivers may have less chance (if any) of going back to school for formal education.

The current study also found that caregivers living in urban areas were less likely to be in food secure households. Therefore, they were more likely to be in both moderate, and severe hunger households than their rural counterparts. This is contrary to what was reported in one study [57], and may be due to differences in contexts and populations. From the program perspective, caregivers are already vulnerable populations who were enrolled in the USAID Kizazi Kipya project because they met at least one project enrollment criterion. As highlighted earlier, the criteria refer to fourteen HIV-related vulnerabilities. Therefore, caregivers living in urban settings may be likely to experience food insecurity because they are, among other things, unable to cultivate their own foods because of lack of land. This is not the case in the rural areas because the caregivers living there have access to land and can grow their own food crops and feed their families even if they do not have adequate money. This implies that, rural and urban caregivers experience food insecurity uniquely, hence a need to target food security interventions to the most disadvantaged urban caregivers.

Although caregivers with mental or physical disability were very few (4%), they were less likely to be in little to no hunger households, hence more likely to experience both moderate

and severe forms of household hunger than their non-disabled counterparts. This was an expected finding and was possible to detect because the study had a very large sample size. Presence of disability implies limited productivity; and since this gets compounded by caregiving obligations, the disabled caregivers are consequently more likely to be food insecure than their non-disabled counterparts. Again, this suggests the need for having appropriate and more specialized strategies for addressing food insecurity among the disabled caregivers to enhance economic, health, social, and overall wellbeing for themselves and the OVC they care for.

Moreover, caregivers with health insurance were more likely to be in food secure households, and less likely to experience moderate and severe forms of household hunger than those who had no health insurance. The possibly underlying reason for this observation is that while the caregivers with health insurance can access health services without having to pay out of pocket (OOP), those without health insurance must incur more OOP monetary expenses for health services and diminish their food budget, and in the end cause food insecurity.

Finally, the current study reveals that, HIV status was associated with food security. Caregivers who were living with HIV and those who did not disclose their HIV status were more likely to be in little to no hunger households, and more likely to be in severe hunger households, but less likely to be in moderate hunger households than those who were HIV negative. The pattern of this observation was not clear, hence a need for further research to explore and explain the relationship.

## Strengths and limitations

Core strengths of this study lie in the large sample size coupled with a national wide geographical coverage that permits generalization of the findings to the whole of Tanzania and beyond, especially in similar areas or contexts. Although some variables such as family size, seasonality, and duration of exposure to the WORH Yetu intervention were unknown, the study design and depth of statistical analyses minimized the possibility of attributing the observed impact of WORTH Yetu on food security on chance or confounding.

## Conclusions

In this study, caregivers' participation in WORTH Yetu was significantly effective in reducing household hunger. The results of the multivariate analysis, adjusted for multiple caregiver and household characteristics, indicate reduced likelihood of being in severe hunger households, and moderate hunger household by 47% and 21%, respectively, and increased the likelihood of food security by 45% because of participation in WORTH Yetu.

At the program level, these findings underscore the need to accelerate coverage of WORTH Yetu along other services as an effective measure towards containment of food insecurity among the caregivers for ultimate wellbeing of the caregivers and OVC. Assessing the extent to which the project has contributed to improved financial resources at household level or nutritional wellbeing of the OVC is worthwhile. Beyond the project, the results can be usefully applied in similar settings and countries to address household hunger.

## Supporting information

**S1 File. Data file for Table 1.**
(XLSB)

**S2 File. Data file for Fig 2.**
(XLSB)

**S3 File. Data file for Fig 3.**
(XLSB)

**S4 File. Data file for Fig 4.**
(XLSB)

**S5 File. Data file for Table 2.**
(XLSB)

## Acknowledgments

Sincere appreciations are extended to the project staff, the consortium partners implementing the USAID Kizazi Kipya project, Civil Society Organizations (CSOs), LCWs, CCWs, District Social Welfare Officers (DSWO), and everyone directly or indirectly involved in the implementation of the USAID Kizazi Kipya project for their precious work.

## Author Contributions

**Conceptualization:** Amon Exavery.

**Data curation:** Amon Exavery, John Charles, Tumainiel Mbwambo, Amal Ally.

**Formal analysis:** Amon Exavery, Shraddha Bajaria.

**Funding acquisition:** Levina Kikoyo.

**Investigation:** Amon Exavery.

**Methodology:** Amon Exavery, John Charles, Shraddha Bajaria, Tumainiel Mbwambo, Amal Ally, Remmy Mseya, Godfrey M. Mubyazi, Levina Kikoyo, Marianna Balampama.

**Project administration:** Levina Kikoyo, Marianna Balampama.

**Resources:** Levina Kikoyo, Marianna Balampama.

**Software:** Tumainiel Mbwambo, Remmy Mseya.

**Supervision:** John Charles, Asheri Barankena, Levina Kikoyo, Marianna Balampama.

**Validation:** Amon Exavery, John Charles, Asheri Barankena, Tumainiel Mbwambo, Amal Ally, Remmy Mseya.

**Visualization:** Amon Exavery, John Charles, Shraddha Bajaria, Amal Ally.

**Writing – original draft:** Amon Exavery, Epifania Minja, Jacob Mulikuza.

**Writing – review & editing:** Amon Exavery, John Charles, Asheri Barankena, Shraddha Bajaria, Epifania Minja, Jacob Mulikuza, Godfrey M. Mubyazi, Levina Kikoyo, Marianna Balampama.

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
