## [Decision Letter · Decision Letter 0]

16 Jun 2021

PONE-D-21-06461

Impact of household economic strengthening intervention on food security among caregivers of orphans and vulnerable children in Tanzania

PLOS ONE

Dear Dr. Exavery,

Thank you for submitting your manuscript to PLOS ONE. After careful consideration, we feel that it has merit but does not fully meet PLOS ONE’s publication criteria as it currently stands. Therefore, we invite you to submit a revised version of the manuscript that addresses the points raised during the review process.

Please use an existing conceptual framework from literature or develop your own conceptual framework to guide your multivariate modelling. Otherwise, it seems a data mining excercise!

We look forward to receiving your revised manuscript.

Kind regards,

Khurshid Alam, Ph. D.

Academic Editor

PLOS ONE

Journal Requirements:

Reviewers' comments:

Reviewer's Responses to Questions

**Comments to the Author**

1. Is the manuscript technically sound, and do the data support the conclusions?

Reviewer #1: Yes

Reviewer #2: Partly

2. Has the statistical analysis been performed appropriately and rigorously? 

Reviewer #1: Yes

Reviewer #2: No

3. Have the authors made all data underlying the findings in their manuscript fully available?

Reviewer #1: Yes

Reviewer #2: No

4. Is the manuscript presented in an intelligible fashion and written in standard English?

Reviewer #1: Yes

Reviewer #2: Yes

5. Review Comments to the Author

Reviewer #1: I read with great interest the paper. I find it well wrote on important focus and setting

below my suggestions

1. Introduction: Use the concept of children at risk to define your vunerable population. The concept of “children at risk” changes worldwide according to each specific context. Africa has a large burden of overall risk factors related to childhood health and development, most of which are of an infective or social origin(The At Risk Child Clinic (ARCC): 3 Years of Health Activities in Support of the Most Vulnerable Children in Beira, Mozambique. Int J Environ Res Public Health). In fact, In the African continent, the greatest risks to children’s health and development are mainly of infective and social origin, particularly exposure to Human Immunodeficiency Virus (HIV), malaria, TB, and malnutrition. It is no coincidence that the scope for child development improvement at the regional level can be identified within three domains: nutrition, environment, and mother-child interaction

2. Methods and results are clear

3. Disucssion: add the role of social determinant of health and discuss better this aspect.

Give some take home message that came from your interesting paper on gloabl health approuch

Reviewer #2: 1. The data analysis and multivariate modelings needs a major review to include more data/information. For instance in Table 3, (n=132,583) is the number of households at baseline, yet in the three models the n=265,166. How did the authors come by that number (365,166) at mid-line? The number at follow-up cannot be more than the number at baseline, else there may be no basis to measure change/impact. Something may be missing here, how many households were interviewed at mid-line and were there the same households interviewed at baseline?. Can the authors show the numbers also in variables ?. Also, we should be comparing the food security situation between those in the intervention (WORTH Yetu) and those who are not in the intervention to identify improvement and not those withing the same group. I recognize that in Figure 1(appendix) there is a comparison of baseline and mid-line with same numbers.But this is not clear because the authors indicated that at mid-line only 10,123 out of the 132,583 households belong to the intervention.

2. Some of the interpretations of the results in Table 3 are wrong. For instance, the interpretation of the Odds Ratios for Rural Vs Urban dwellers (Line 348 - 350) is wrong, it should be the opposite. "While caregivers in rural areas were less likely to be in little to no hunger households, they were more likely to be in both moderate and severe hunger households, than their urban counterparts". They authors however interpreted it correctly in the discussion section (lines 427-430

3. There are Tables 1 and 3, but no table 2 is shown throughout the manuscript

4. Table 1 showed be presented including the data of the mid-line. That is, show baseline and Mid-line side-by-side

5. The Authors indicated the study design is longitudinal: baseline data in 2017/18 and mid-line data in 2019. When exact (month) did the baseline end? and when exactly (period/month) in 2019 was the mid-line data collection done? This information is important because it has implication for the results and conclusions drawn

We are told the intervention began in 2019, when will the intervention end?

5. The authors should provide more details about the study design and data collection including number interviewed at mid line etc.

6. PLOS authors have the option to publish the peer review history of their article (what does this mean?). If published, this will include your full peer review and any attached files.

Reviewer #1: **Yes: **Di Gennaro Francesco

Reviewer #2: No

---

## [Author Response · Author response to Decision Letter 0]

22 Jul 2021

See "Response to Reviewers" file

---

## [Decision Letter · Decision Letter 1]

9 Feb 2022

Impact of household economic strengthening intervention on food security among caregivers of orphans and vulnerable children in Tanzania

PONE-D-21-06461R1

Dear Dr. Exavery,

We’re pleased to inform you that your manuscript has been judged scientifically suitable for publication and will be formally accepted for publication once it meets all outstanding technical requirements.

Kind regards,

Khurshid Alam, Ph. D.

Academic Editor

PLOS ONE

Additional Editor Comments (optional):

Reviewers' comments:

Reviewer's Responses to Questions

**Comments to the Author**

1. If the authors have adequately addressed your comments raised in a previous round of review and you feel that this manuscript is now acceptable for publication, you may indicate that here to bypass the “Comments to the Author” section, enter your conflict of interest statement in the “Confidential to Editor” section, and submit your "Accept" recommendation.

Reviewer #1: All comments have been addressed

2. Is the manuscript technically sound, and do the data support the conclusions?

Reviewer #1: Yes

3. Has the statistical analysis been performed appropriately and rigorously? 

Reviewer #1: Yes

4. Have the authors made all data underlying the findings in their manuscript fully available?

Reviewer #1: Yes

5. Is the manuscript presented in an intelligible fashion and written in standard English?

Reviewer #1: Yes

6. Review Comments to the Author

Reviewer #1: authors improved their paper that now can be accept. I appreciate the paper, the research question and the setting of study

7. PLOS authors have the option to publish the peer review history of their article (what does this mean?). If published, this will include your full peer review and any attached files.

Reviewer #1: No

---

## [Editor Report · Acceptance letter]

15 Feb 2022

PONE-D-21-06461R1 

Impact of household economic strengthening intervention on food security among caregivers of orphans and vulnerable children in Tanzania 

Dear Dr. Exavery:

I'm pleased to inform you that your manuscript has been deemed suitable for publication in PLOS ONE. Congratulations! Your manuscript is now with our production department. 

Kind regards, 

on behalf of

Dr. Khurshid Alam 

Academic Editor

PLOS ONE